# Gene Ontology (GO)-Driven Inference of Candidate Proteomic Markers Associated with Muscle Atrophy Conditions

**DOI:** 10.3390/molecules27175514

**Published:** 2022-08-27

**Authors:** Angelique Stalmach, Ines Boehm, Marco Fernandes, Alison Rutter, Richard J. E. Skipworth, Holger Husi

**Affiliations:** 1Centre for Health Science, Division of Biomedical Sciences, Institute of Health Research and Innovation, School of Health, Social Care and Life Sciences, University of the Highlands and Islands, Inverness IV2 3JH, UK; 2Edinburgh Cancer Research UK Tissue Group, Institute of Genetics and Cancer, University of Edinburgh, Edinburgh EH4 2XR, UK; 3Clinical Surgery, University of Edinburgh, Royal Infirmary of Edinburgh, Edinburgh EH16 4SA, UK; 4Biozentrum, University of Basel, 4056 Basel, Switzerland; 5Department of Psychiatry, University of Oxford, Oxford OX3 7JX, UK; 6Institute of Cardiovascular and Medical Sciences, University of Glasgow, Glasgow G12 8TA, UK

**Keywords:** proteomics, muscle wasting, sarcopenia, cancer cachexia, biomarker

## Abstract

Skeletal muscle homeostasis is essential for the maintenance of a healthy and active lifestyle. Imbalance in muscle homeostasis has significant consequences such as atrophy, loss of muscle mass, and progressive loss of functions. Aging-related muscle wasting, sarcopenia, and atrophy as a consequence of disease, such as cachexia, reduce the quality of life, increase morbidity and result in an overall poor prognosis. Investigating the muscle proteome related to muscle atrophy diseases has a great potential for diagnostic medicine to identify (i) potential protein biomarkers, and (ii) biological processes and functions common or unique to muscle wasting, cachexia, sarcopenia, and aging alone. We conducted a meta-analysis using gene ontology (GO) analysis of 24 human proteomic studies using tissue samples (skeletal muscle and adipose biopsies) and/or biofluids (serum, plasma, urine). Whilst there were few similarities in protein directionality across studies, biological processes common to conditions were identified. Here we demonstrate that the GO analysis of published human proteomics data can identify processes not revealed by single studies. We recommend the integration of proteomics data from tissue samples and biofluids to yield a comprehensive overview of the human skeletal muscle proteome. This will facilitate the identification of biomarkers and potential pathways of muscle-wasting conditions for use in clinics.

## 1. Introduction

Skeletal muscle amounts to ~40% of the human body weight and entails 50–75% of all body proteins [1,2]. Moreover, skeletal muscle tissue has a complex tissue architecture consisting of a multitude of cell types: immune cells, endothelial cells, fibro-adipogenic progenitors (FAPs), tenocytes, satellite cells, muscle fibers, and neural cells [3]. These cells communicate amongst each other in a paracrine fashion, and skeletal muscle itself has also been shown to communicate to other organs and tissues (i.e., adipose tissue) in an endocrine fashion through the secretion of so-called myokines by muscle contraction [4]. Conversely, adipose tissue may also play an active role in cancer-associated cachexia, whereby lipolysis has been shown to be permissive of skeletal muscle wasting, and an essential part in the pathogenesis of cancer-associated cachexia [5]. It is, therefore, critical to sustain skeletal muscle function and integrity across the lifespan to retain quality of life and prevent comorbidities associated with muscle atrophy and loss of function. Skeletal muscle strongly relies on metabolic homeostasis, a delicate balance between catabolic and anabolic pathways. A shift to either catabolism (protein breakdown) or anabolism (protein build-up) would generally result in atrophy or hypertrophy, respectively. In diseases of muscle wasting, one can often observe a reduction in anabolism, e.g., due to lack of nutrients and immobility, and increased catabolism, e.g., as a response to metabolic stress and increased autophagy [6,7]. Muscle atrophy is a prevalent diagnosis in clinic and accompanies a variety of pathologies. The most common diseases associated with muscle atrophy—that will likely affect most of us at some stage in our lives—are sarcopenia, (the aging-related loss of muscle mass and function [8,9]), cachexia, (the disease, e.g., cancer, associated with a loss of muscle mass, with or without fat loss, and a progressive loss of function [10]), and muscle wasting (induced by, e.g., inactivity or bed-rest [2]).

Currently, there are no effective treatments for muscle wasting on the market. Whilst some drugs in development have resulted in increased muscle mass, no treatment to date has managed to improve functional outcome measures [11]. The only effective treatment to date is exercise. Whilst exercise has shown preventative effects and the potential to reduce the risk of falls in elderly [12], immobile or bedridden patients cannot benefit, thus creating the unmet need to identify (i) new treatments (i.e., exercise mimetics), and (ii) early biomarkers of muscle wasting and loss of function, in order to initiate preventative measures before onset. A plethora of new therapies targeting a variety of molecular mechanisms are already under development [11], and new druggable pathways are consistently being identified with hope for a more stratified approach in future patient care. Whilst there is a variety of validated biomarkers in patients for neuromuscular disorders such as Duchenne muscular dystrophy (serum miR-499) [13], or Charcot-Marie-Tooth disease (NCAM1 and GDF15) [14], there is a lack of early biomarkers that could predict muscle-wasting diseases such as sarcopenia or cachexia. Some biomarkers associated with physical frailty and sarcopenia have been identified, such as gut bacteria and markers of inflammation [15], as well as C-reactive protein (CRP) [15,16], also a marker of muscle wasting in cancer cachexia [17], but longitudinal patient studies are yet to be undertaken to evaluate the potential of these biomarkers to predict onset, treatment response and the progression of such muscle wasting disease. Although genome sequencing has uncovered a great deal of understanding about disease mechanisms, these do not always translate into a one-to-one ratio in protein levels. Despite the known benefit and great promise of protein biomarker discovery, in particular from less invasive biopsies such as biofluids i.e., serum, plasma and non-invasive urine samples in comparison to invasive tissue biopsies, there are still only a limited number of protein biomarkers that are actively used in the clinic [18,19]. Therefore, the knowledge that skeletal muscle tissue plays a significant role in endocrine signaling, in particular upon insult, warrants the study of protein biomarkers in not only skeletal muscle, but also tissue affected by secretion of such myokines: biofluids.

The capacity of skeletal muscle to secrete myokines gives biofluids enormous prognostic power since proteomic biomarkers could potentially predict disease states at timepoints shortly after secretion, before the body has had enough time to enact physiological change. Likewise, the known para- and endocrine role that adipo- and myokines play in the regulation of each other could prove useful in biomarker discovery [20]. One should note that osteokines also strongly contribute to this signaling triad between muscle, adipose, and bone, however, are not easily used in biopsies.

Given the clinical relevance of protein biomarkers for clinical diagnostics and precision medicine, we set out to identify the body of literature that conducted proteomic studies investigating suitable tissue (skeletal muscle and adipose tissue) and biofluids (serum, plasma and urine) in four muscle atrophy conditions: sarcopenia, cachexia, muscle wasting, and aging itself. We conducted a meta-analysis to condense (i) biological processes and functions common or unique to the four conditions, and (ii) potential protein biomarkers. To do so, we performed a gene ontology (GO) analysis of 24 human proteomic studies and found that the great heterogeneity in study design, weight-loss definition, and patient characteristics contributed to little overlap of up- and/or downregulated proteins identified across studies of the same muscle atrophy condition. Whilst individual proteins lacked commonality in direction across studies and conditions, all four atrophy groups shared common processes that were up- and downregulated. In tissue samples, commonalities such as sarcomere organization, muscle filament sliding, maintenance of mitochondrion location, skeletal muscle tissue regeneration, negative regulation of calcium-mediated signaling, short-chain fatty acid metabolic process were found; and in biofluid samples, only muscle cell migration was shared. We highlight that this approach can uncover common biological processes in muscle wasting that are not revealed by single studies and therefore demonstrate the strength of integrating multiple datasets. Furthermore, we stress the need for the standardization of protocols—in clinical and research settings—and robust quantitative methods for data analysis, and most importantly, appropriate patient selection, including both sexes.

## 2. Results

### 2.1. Overview of Studies Included

The 24 studies included in the analysis were peer-reviewed original articles with publication dates ranging between 2010 and 2021 (Table 1). One study covered cachexia and sarcopenia and was split accordingly in the study of biological processes and function analysis. Studies reported differentially expressed proteins associated with cachexia (*n* = 9), sarcopenia (*n* = 4), aging (*n* = 7) and muscle-wasting conditions (*n* = 5) in human volunteers, with the total numbers of participants in each study spanning between N = 2 and N = 240. Protein identification was carried out either in muscle biopsies (*n* = 15) or biofluids (*n* = 10), defined as serum (*n* = 5), plasma (*n* = 2), or urine samples (*n* = 3). One study was conducted on adipose tissue. A large proportion of studies (*n* = 21) used mass spectrometry-based proteomics to identify and quantify proteins in the various samples. Details of the studies included are summarized in Appendix A.

### 2.2. Analysis at the Molecular Level

Based on the extracted data from individual studies, a wide range of significantly up- and downregulated proteins were identified in tissue biopsies and/or biofluid samples across the various conditions under study, with a total number of proteins ranging between *n* = 4 in biofluid samples associated with sarcopenia and *n* = 391 in tissue samples associated with cachexia (Table 2 and a complete list of molecules in Appendix A).

Overall, there was little overlap at the protein level across the four muscle atrophy conditions (Figure 1 and Appendix A). The largest proportion of unique, non-overlapping up- and/or downregulated proteins were observed in cachexia, with 87% (340/391) of proteins in tissue samples and 100% (97/97) in biofluid samples being identified exclusively in patients with cachexia, but not in individuals presenting with other muscle wasting-related conditions. Across all four muscle atrophy conditions, only two proteins, ENO3 and CKM, were commonly found in tissue samples, while no proteins were common to all conditions in biofluid samples. The proportions of unique proteins associated with muscle wasting, sarcopenia, and aging were 15% (3/20), 22% (9/41), and 50% (56/113) in tissue samples (Figure 1A). In contrast, almost all the proteins identified in biofluid samples were uniquely associated with their respective conditions (85% (22/26), 100% (4/4), and 86% (32/37)) (Figure 1B).

Across proteomics studies for each of the four muscle-wasting conditions, very little protein overlap was observed between tissue and biofluid samples (Table 3). The proportion of overlapping proteins for muscle wasting, cachexia, sarcopenia, and aging in tissue/biofluid samples was 0/3.8%, 5.4/1.0%, 7.3/0%, and 30.1/0% respectively, and the proportion of overlapping proteins with the same directionality was even less (Table 2).

In our current analysis, possible factors that might explain the low proportion of overlapping proteins across studies include the many types of cancer (six different types across nine studies), and the heterogenous definition of cachexia in the individual studies. Similarly, heterogeneity in the definition of weight loss and patient characteristics across studies might partly explain the low proportion of overlapping proteins with the same directionality in sarcopenia, muscle wasting and aging-related muscle atrophy conditions. Additionally, the different methodologies observed across studies in relation to protein extraction and sample processing, analytical and quantitation approaches, and the mass spectrometry platform (see in Appendix A) may lead to the identification of a wide range of different proteins. Altogether, the lack of a standardized approach for generating proteomics data may contribute to a low overlap of molecules across studies.

Due to the limited overlap of proteins with the same directionality, a meta-analysis focusing on differences observed at the molecular level was not feasible. However, this does not preclude the existence of convergent molecular functions and pathways involved in biological processes affected by these conditions. Therefore, we performed a GO enrichment analysis via over-representation statistical methods for each muscle atrophy-related condition, producing a set of enriched GO terms to describe changes affecting biological processes and functions.

### 2.3. Differences and Convergent Molecular Pathways across Cachexia, Sarcopenia, Aging, and Muscle-Wasting Conditions

Clustering analysis revealed overlapping functions common to all four atrophy conditions: skeletal adaptation, phosphagen metabolic process, proton-transporting ATP synthase activity, negative regulation of calcium-mediated signaling, and oxaloacetate metabolic process accounted for 64% of the GO terms in tissue samples (Figure 2A and Appendix A), and muscle cell migration (100% of the GO terms) in biofluid samples (Figure 2B and Appendix A).

The convergent biological processes and function across all four atrophy conditions center around seven protein-driven GO clusters that can be described as skeletal mass and function, oxidative stress response, energy transfer, phosphagen system, glycolytic system, oxidative (mitochondrial) system (tissue samples, Figure 2A), and muscle tissue repair (biofluid samples, Figure 2B). Indeed, different physio-pathologies that result in muscle atrophy, such as age-related sarcopenia and cancer-induced cachexia, have been shown to exhibit similar molecular, cellular, and systemic effects characterized by impaired protein homeostasis, metabolic remodeling, impaired mitochondrial function, increased oxidative stress response, impaired energy expenditure, and altered muscle composition, leading to muscle wasting [45,46,47,48]. Our analysis therefore shows that changes in muscle adaptation and metabolic processes might be an indicator of muscle damage and/or remodeling underlying all four atrophy conditions. Muscle-wasting conditions present similarities in their clinical manifestations and can occur concurrently in the same patient [49]; however, they do have distinct etiologies, and factors that promote cachexia are known to be different from those involved in sarcopenia, with inflammation being one of the key features in cachexia [50,51,52]. This suggests that specific protein-driven GO terms may be associated with each muscle atrophy condition. We next examined the proteomic changes associated with atrophy induced by cachexia, sarcopenia, aging, and muscle wasting. Our analysis suggested that distinguishable biological processes and functions were affected in the different conditions under study, as summarized in the subsequent sections.

### 2.4. Analysis of the Biological Processes and Functions

#### 2.4.1. Biological Processes and Functions Affected in Cachexia

In total, 247 upregulated and 236 downregulated differentially expressed proteins (reported as gene names) were identified across cachexia-related studies: 212 up- and 175 downregulated proteins were identified in tissue (Figure 3A), and 35 up- and 61 downregulated proteins were identified in biofluid (Figure 3B). In general, different proteins were found in the two types of samples (muscle biopsies and biofluid), resulting in different enriched pathways. In muscle tissue, lactate metabolic process, NADH oxidation, and glycolytic process were the major enriched pathways for the upregulated genes. In contrast, antimicrobial humoral response, regulation of ATPase activity and cellular response to interleukin-12 were the major enriched pathways for the downregulated genes. In biofluid samples, negative regulation of cysteine-type endopeptidase activity involved in apoptotic processes and acute-phase responses were the major enriched pathways associated with the upregulation of genes. Phospholipid catabolic processes, calcium-dependent protein kinase activity and death receptor activity were the main pathways associated with downregulation in biofluid.

Overall, there was little overlap of proteins identified across cachexia studies, with 0.0% (0/97) of proteins overlapping in biofluid samples and 4.3% (17/391) of proteins in tissue samples being commonly identified with the same directionality (fold-change or up-/downregulation) (Table 2). In the case of cachexia, this lack of protein overlap across studies is not surprising considering the complexity and multifactorial characteristics of the syndrome [53,54]. A wide range of potential biomarkers involved in the muscle wasting process has been identified and reviewed elsewhere [55,56]. Many candidate biomarkers, alone or in combination, might serve as biomarkers of cancer cachexia, including proteins, miRNAs, and metabolites. However, inconsistencies across a range of cancer types, reliable detection methods, and other confounding factors have been reported and represent many challenges that prevent their validation and implementation in the clinic.

Based on our analysis, the altered pathways involved in cachexia are all known hallmarks of metabolic alterations in cancer, controlled by tumor–host-cell interactions, key oncogenes, tumor suppressors, and other regulatory molecules that characterize the uncontrolled growth and proliferation of cancer cells [57]. The altered metabolism of cancerous cells and tissue induces systemic changes of the host body metabolism by secreting humoral factors (i.e., TNF-α, IL-1 and IL-6) and pro-cachectic factors (i.e., proteolysis-inducing factor and lipid mobilization factor) that lead to a generalized catabolic state followed by significant and progressive energy loss from the host’s tissue [54,58]. Interestingly, the upregulation of lactate metabolic process and NADH oxidation identified as major enriched pathways in our analysis would support the ‘reverse Warburg effect’ paradigm, in which lactate plays an active role in carcinogenesis [59,60,61] and would also suggest a link between lactogenesis and cachexia. Using a model of cachectic myotubes, Mannelli et al. demonstrated that cachexia was induced in healthy myotubes by a metabolic shift from oxidative to fermentative metabolism due to altered mitochondria, mediated by IL-6 in association with an increased lactate formation and decreased oxygen consumption observed in the cachectic myotubes [62]. The complexity of interacting pathways involved in cachexia, as indicated in our analysis, supports prior evidence of a cancer-induced systemic pathogenic network, suggesting that cachexia results from the systemic metabolic reprogramming of host cells via host–tumor-cell crosstalk, promoting primary and metastatic tumor growth [54,58,63].

To date, very few studies have investigated the proteomics changes related to muscle wasting in cancer patients and whether common proteome modifications that characterize muscle atrophy can be observed across different pathologies. Aniort et al. used LC-MS/MS and functional enrichment analysis to investigate the proteome from muscle biopsies of patients with early-stage lung cancer (*n* = 7), patients with chronic hemodialysis (*n* = 7), and healthy patients as a control (*n* = 7), to investigate whether modifications are commonly observed in conditions known to induce muscle atrophy [23]. They found that activation of the Wnt/β-catenin pathway and the overexpression of several enzymes from the ubiquitin-proteasome system (namely E3 ligases) as well as autophagy-related enzymes (such as cathepsin L and sequestosome 1) were commonly observed in patients with cancer or kidney failure, suggesting that the degradation of the muscle contractile function and the cytoskeleton integrity may be similar, irrespective of the underlying pathological condition. A study by Ebhardt et al. compared the soluble proteome of muscle biopsies from cancer cachexia patients (*n* = 5) to that from cancer patients without cachexia (*n* = 14) and healthy individuals with (*n* = 8) and without (*n* = 10) age-related muscle loss [21]. Proteome analysis by LC/MS-MS followed by a protein-protein interaction analysis identified 16 highly regulated proteins with statistical significance, clustered around three modules, namely the Fo complex, electron transfer chain, and contractile fiber, distinguishing cachexia from other patients or healthy elderly individuals. An in vitro time-lapse experiment mimicking cancer cachexia using human skeletal muscle cells from an 83-year-old donor identified changes in phosphorylation levels and perturbation of the FAK signal transduction pathway governing cell fusion that plays a critical role in preventing muscle regeneration [21].

Besides interrogating the muscle proteome to discover potential mechanisms of action and therapeutic targets, several studies (10 out of 24) have profiled the proteome of the plasma, serum and/or urine samples of patients to discover novel proteomics biomarkers that could potentially identify at-risk patients or stage muscle wasting severity in a non-invasive manner.

In a study of patients with newly diagnosed or suspected gastrointestinal cancer, with (*n* = 32) or without (*n* = 27) cachexia, the plasma proteome was profiled using 760 antibodies (targeting 698 individual proteins) [29]. Of the six proteins that showed significant differences between the patients with and without cachexia, three were lower (CNPD1, APOA4, DACH1) while three were higher (BCL3 NARS2, ATP13A4) in cachexia. Reduced CNPD1 levels measured in a validation cohort of patients with advanced pancreatic cancer were significantly correlated with the percentage of weight loss and fat oxidation, whereas a significant positive correlation was observed between CNDP1 and survival (measured as days of survival following diagnosis) [29].

Urine has a rich protein content and has been used in research and clinical settings for the discovery of novel diagnostic or prognostic biomarkers in specific disease states, using mass-spectrometry-based approaches [64].

Skipworth et al. used 1D gel electrophoresis and matrix-assisted laser desorption/ionization, and liquid chromatography tandem mass spectrometry to compare the urinary proteome of patients with gastro-esophageal cancer cachexia (*n* = 8), weight-stable gastro-esophageal cancer (*n* = 8), and healthy controls (*n* = 8) [28]. They found a greater number of urinary proteins in cachectic samples (*n* = 199) when compared with weight-stable (*n* = 79) and control samples (*n* = 49), with proteins specific to cachectic samples including myosin species (muscle origin), α-spectrin and nischarin (cytoskeletal), and microtubule-associated proteins (microtubule–actin crosslinking factor; microtubule-associated protein-1B; bullous pemphigoid antigen 1) [28]. Such potential biomarkers deserve further validation in larger cohorts to better define groups of patients at risk of worsened outcomes.

Another mass spectrometry-based technique employed to identify potential urinary biomarkers is surface-enhanced laser desorption/ionization; together with model building using decision tree analysis, Husi et al. identified prospective biomarkers of myosteatosis [44] and dynapenia (the age-associated loss of muscle strength) [42] by comparing the expression profiles of potential biomarker peaks between groups, to establish a proteomic fingerprint pattern which, upon validation, can be used in clinical diagnostics. Such an approach not only allows stratification and pattern matching to search for specific cachexia-related biomarkers in the urine of cancer patients, but also identifies key protein fragments that may be related to the pathophysiology of muscle wasting in cancer. For example, the downregulation of fragments of CTSC, agrin, ARSA and GFAP was observed in patients with upper gastrointestinal tract cancer and myosteatotic vs. non-myosteatotic [44], whilst downregulation of annexin A1 and COL15A1 chain was observed in patients with a mixture of cancers and with leg-power measurement-based dynapenia [42].

#### 2.4.2. Biological Processes and Functions Affected in Sarcopenia

Across sarcopenia studies, 30 up- and 11 downregulated genes were identified in tissue biopsies (Figure 4A), and three up- and one downregulated gene were identified in biofluid samples (Figure 4B). In contrast to the complexity of interacting pathways involved in cancer-related cachexia, the key changes associated with sarcopenia included the downregulation of pathways related to energy metabolism, namely the phosphagen metabolic system, gluconeogenesis pathway, and the proton-transporting ATP synthase activity, among others (Figure 4A). In a study investigating the role of dysregulated intracellular creatine metabolism in disuse muscle, Luo et al. demonstrated that seven days of unilateral limb immobilization resulted in greater intracellular creatinine transporter and reduced phosphocreatine content when compared with the non-immobilized control limb in a cohort of 15 healthy men; they hypothesized that the lower intramuscular phosphocreatine content observed might be explained by the significantly lower fast-twitch fiber cross-sectional area compared with the control biopsies from the non-immobilized limb [65]. Sarcopenia and aging have been associated with the preferential atrophy of fast-twitch fibers and a glycolytic-to-oxidative metabolic shift [39,66]. In two recent studies conducted on older adults, individuals presenting with a hip fracture also showed extensive Type II (fast-twitch) muscle fiber atrophy [67,68]. Whilst glycolytic muscle fibers are responsible for rapid muscle force production during muscle contraction and therefore have an essential role in facilitating strength, they also fatigue quicker as their glycogen storage depletes during glycolysis. Whilst oxidative fibers rely on aerobic ATP synthesis from a dense network of mitochondria, they fatigue less quickly and maintain posture and prevent falls and fractures. Thus, it makes sense that a disturbance in pathways associated with mitochondrial function, or glucose metabolism, could contribute to the loss of muscle force that accompanies sarcopenia.

Another important consideration in identifying the key pathways associated with sarcopenia is the impact of gender on the muscle proteome. Bergen III et al. proposed a differential role for myostatin, a negative regulator of skeletal muscle mass, as a potential contributor to sarcopenia in women and a homeostatic regulator of muscle mass in men. Through a targeted proteomics analysis using liquid chromatography with tandem mass spectrometry measuring circulating concentrations of myostatin, the authors showed that relative to lean mass both older and sarcopenic older women had >23% higher myostatin levels than younger women, whereas younger men had >25% higher myostatin concentrations than older men with and without sarcopenia. In both sexes, the highest concentrations of follistatin-related gene protein, a key myostatin inhibitor, were observed in the sarcopenic older group [31].

#### 2.4.3. Biological Processes and Functions Affected by Aging

Across studies investigating the effect of age on changes to the muscle proteome, 52 upregulated and 43 downregulated differentially expressed genes were identified (Figure 5A), and 33 up- and 4 downregulated proteins were identified in biofluid (Figure 5B). In muscle tissue, the protein-folding chaperone, fructose 1,6-bisphosphate metabolic process, and glutathione peroxidase activity were the major enriched pathways for the upregulated genes. Responses to fatty acid, the tricarboxylic acid cycle, and cellular aldehyde metabolic process were the major enriched pathways for the downregulated genes. Enriched pathways associated with aging in biofluid (serum samples) came from a single study of frail patients versus non-frail subjects. The main upregulated pathways identified were complement activation (alternative pathway), the regulation of humoral immune response, and acute-phase response.

Overall, age-related changes in protein expression included more efficient quality control and folding mechanisms, supported by an upregulation of the protein-folding chaperone, muscle contraction, energy metabolism, ion transport, and cellular oxidative activities in tissue biopsies of older vs. younger individuals (Figure 5A).

An increase in oxidative modification (also known as carbonylation) of proteins has also been featured across a number of studies. This is a known hallmark of aging and age-related diseases, with the accumulation of modification products likely contributing to abnormal cellular functions [69,70]. Lourenço dos Santos et al. identified 17 proteins in the *rectus abdominis* muscle that were increasingly carbonylated in older (52–76 years) vs. younger (0–12 years) healthy men, all involved in muscle contraction (myosin 7, troponin T, myosin-binding protein C, and LIM domain-binding protein 3), energy metabolism (fructose-bisphosphate aldolase A and glycogen phosphorylase), and energy transduction (Creatine kinase M-type), suggesting that oxidative stress associated with aging targets proteins involved in contractile activity, as well as structural and regulatory processes [36]. Additionally, the authors identified increased carbonylation of heat shock 70 kDa protein (HSP70) in the older vs. younger group. HSP70s are highly conserved proteins known to function as intracellular molecular chaperones to protect the proteome by folding denatured polypeptides, promoting the degradation of severely damaged proteins, and protecting cells against damage-induced stimuli [71,72]. During aging, increased protein damage may be exacerbated by a declining heat shock response, reduced levels of heat shock proteins, and the resultant loss of protein quality control, thereby preventing repair of protein damage, leading to degeneration and cell death [72]. Additionally, a proteomics analysis of cellular protein extracts from young and senescent human myoblasts showed that key proteins involved in carbohydrate metabolism, cellular morphology, migration and proliferation, as well as protein quality control, protein degradation and free radical scavenging, were increasingly carbonylated and/or modified with advanced glycation/lipid peroxidation end products in the senescent when compared with the young myoblasts. The authors also identified six enzymes (namely aldolase, triosephosphate isomerase, glyceraldehyde 3-phosphate dehydrogenase, phosphoglycerate mutase, enolase, and pyruvate kinase) involved in glycolysis to be increasingly modified in the senescent myoblasts resulting in a metabolic shift, observed likely due to an impairment in glycolysis and/or tricarboxylic acid (TCA) cycle [73].

While the impact of sex and hormones on muscle wasting is well known and has been reviewed elsewhere [74,75], the molecular mechanisms underpinning loss in muscle function associated with hormonal status, independent of age, are scarcely known. A study investigating the proteomic profiles of muscle samples from 24 pre- and postmenopausal women was conducted to establish the molecular differences associated with age (30–34 years old vs. 54–62 years old), menopausal status (premenopausal vs. postmenopausal), and the use of hormone replacement therapy (HRT; user vs. nonuser) [76]. Of the 797 proteins quantified using label-free proteomic analysis, 79 differentially expressed proteins were associated with age, independent of HRT use, while 17 novel proteins were differentially expressed in postmenopausal women without HRT and 49 proteins were associated with HRT in comparison to premenopausal women. The major canonical pathways associated with different protein expression included: mitochondrial dysfunction, oxidative phosphorylation, glycolysis, and the TCA-cycle—all strong indicators for affected energy metabolism. Major biological processes predicted to be affected were related to: cell death, apoptosis, cell survival, contractility of the muscle and glycolysis. Estrogen hormones have been shown to influence the binding of myosin heavy chain to actin through phosphorylation of the regulatory light chain (RLC), thereby preserving muscle mass and quality of the contractile proteins in females [77]. Miller et al. showed that in comparison to muscle biopsies from young individuals, the phosphorylation of RLC was reduced in fibers of old, estrogen-deficient women but not in old men, indicative of an impact of estrogens on the post-translational modification of myosin RLC [78]. In contrast, proteomics analysis of single muscle fibers in physically active elderly (70.9 years old) and young (23.0 years old) men revealed that post-translational modifications including the phosphorylation of myosin light chain 2 slow and oxidation of myosin heavy chain increased with aging. Such qualitative adaptations rather than quantitative changes in protein content are likely to be involved in men’s age-related impairment of muscle fiber structure and function [34].

#### 2.4.4. Biological Processes and Functions Affected by Muscle Wasting

Across studies investigating the proteome of muscle wasting, genes were largely downregulated in patients affected by the condition: in tissue biopsies, 15 genes were downregulated (only five genes were upregulated), with the main pathways involved being aerobic respiration and oxidative phosphorylation (Figure 6A). In biofluid samples, 21 genes were associated with the downregulation of renal absorption, the negative regulation of actin filament depolymerization, dipeptidyl–peptidase activity, and microglial cell activation (Figure 6B).

In tissue samples, similar to the aging process, muscle wasting was associated with mitochondrial dysfunction, characterized by the downregulation of oxidative phosphorylation and aerobic respiration. Analysis of the biofluid proteome highlighted different downregulated functions related mainly to cell activation, migration, and regeneration. Interestingly, our analysis revealed reduced dipeptidyl–peptidase activity in muscle wasting. Inhibition through administration of a dipeptidyl–peptidase-4 inhibitor has shown to result in anabolic action on skeletal muscle in a mouse model [79] as well as an increase in muscle mass and muscle/fat ratio in humans [80]. However, the mechanisms by which dipeptidyl–peptidase activity exerts action on muscle remains unclear [81].

## 3. Methods

### 3.1. Study Selection and Inclusion Criteria

We conducted a gene ontology analysis to identify potential biomarkers and investigate pathways involved in muscle atrophy associated with muscle wasting, cachexia, sarcopenia, and aging. We used PubMed, SCOPUS and Clarivate/Web of Science to conduct a search of original research articles published in the last 15 years (from 2006 through to 15 November 2021) and conducted in the English language. Keyword search terms included (“cachexia” OR “sarcopenia” OR “muscle wasting”) AND “proteomics” for PubMed and SCOPUS, and (“sarcopenia proteomic*” OR “cachexia proteomic*” OR “wasting proteomic*”) for Clarivate. Both SCOPUS and Clarivate searches were also restricted to the MESH term “human”. The review was limited to clinical trials, controlled clinical trials, journal articles, meta-analyses, and randomized controlled trials that reported proteomics data in human participants. Additionally, we also conducted an open web search using “Sarcopenia proteomic biomarker” as keyword search terms.

Following the initial data search and removal of duplicate articles, titles and/or abstracts were manually screened to identify studies that met the inclusion criteria, and full texts were subsequently retrieved to confirm eligibility. The study selection process is shown in Figure 7.

### 3.2. Exclusion Criteria

Animal and in vitro or cell culture studies were excluded. Selected articles had to include proteomics data, and reports that focused on other -omics were excluded if they did not include proteomics. Studies focusing on muscle atrophy associated with certain genetic disorders such as Duchenne Muscular Dystrophy, or Spinal Muscular Atrophy were also excluded.

### 3.3. Data Collection

Studies included in the review were extracted into a tabulated format. The following data were extracted: the condition associated with muscle atrophy (whether cachexia, sarcopenia, aging, or muscle wasting), definition of weight loss (if provided in the methodology of the individual studies), total number of volunteers or patients included and number per group, cancer type (in the case of cachexia), tissue in which proteomics analyses were carried out, list of proteins (including fold-change, *p*-value, directionality), and the analytical method used to carry out the proteomics analyses. Outcome measures included proteins identified (listed as gene name) that were differentially expressed in the study groups (*p*-value and/or false discovery rate, depending on individual studies’ definition of differential expression) and fold change, log-transformed fold change, or directionality (up- or downregulation in cases where quantification was not reported). Data extracted were harmonized to UniProt and SwissProt identifiers (human) and split into two sets each: all molecules across all studies per atrophy group and split into up- or downregulated molecules.

### 3.4. Gene Ontology Analysis

Gene ontology (GO) analysis was carried out using ClueGO (v2.5.8) and CluePedia (v1.5.8) running in Cytoscape (v3.9.0) and Java (v11). The UniProt/SwissProt reference database (accessed on 10 December 2021) was used, with a compiled GO reference file (accessed on 13 May 2021). Venn diagrams comparing the overlapping of proteins identified across the four conditions were generated using the Venn Diagrams webtool (https://bioinformatics.psb.ugent.be/webtools/Venn/, accessed on 31 January 2022).

## 4. Conclusions and Future Directions

Our current analysis aimed to identify biological processes and functions impacted by muscle wasting and revealed distinguishable pathways associated with muscle atrophy, whether related to aging, sarcopenia, muscle wasting, or cachexia. This GO analysis based on published data from proteomics studies of muscle atrophy conditions can be used to (i) identify the potential mediators of muscle atrophy and (ii) cross-compare different muscle wasting conditions from individual proteomics studies. We observed very little overlap of proteins with the same directionality across studies and hypothesize that differences in the weight-loss definition, patient characteristics, mass spectrometry-based methods, tissue types, and cancer types could partially explain this heterogeneity at the molecular level. Common biological processes and functions across all four atrophy conditions pertained to skeletal mass and function, oxidative stress response, energy transfer, phosphagen/glycolytic/oxidative systems, and muscle tissue repair. Changes in muscle adaptation and metabolic processes might therefore be mechanisms associated with muscle damage and/or remodeling common to all four atrophy conditions. Additionally, our analysis identified distinct biological processes and functions associated with each muscle atrophy condition. The major biological processes and functions impacted in cachexia appeared to largely relate to changes in host metabolism, with lactate metabolic process, NADH oxidation, and glycolytic process identified as the major enriched pathways for the upregulated genes, presumably driven by a metabolic reprogramming of host cells via host–tumor-cell crosstalk. By contrast, key changes associated with sarcopenia included the downregulation of pathways related to energy metabolism, indicative of a preferential atrophy of fast-twitch fibers and a glycolytic-to-oxidative metabolic shift that could contribute to the loss of muscle force. In aging, the upregulation of protein folding chaperone, muscle contraction, energy metabolism, and cellular oxidative activities, together with an impaired mitochondrial capacity observed in tissue biopsies of older vs. younger individuals, are known hallmarks of cellular senescence promoting the loss of protein quality control, thereby preventing repair of protein damage, leading to degeneration and cell death. Similar to the aging process, muscle wasting was associated with mitochondrial dysfunction, and impaired cell activation, migration, and regeneration, possibly indicative of an impaired cellular anabolism. Altogether, our analysis revealed common and specific biological processes and functions across various muscle atrophy conditions which, despite their distinct etiologies, often present with similar manifestations and poor patient outcomes. At the molecular level, each of the muscle atrophy conditions is distinguishable from one another, driven by different progenitor events, which brings the possibility to develop targeted treatment and improve overall outcomes.

Other factors known to impact muscle mass include sex, hormonal status, and/or fat infiltration. The small number of studies did not allow us to conduct our analysis using stratification parameters. As the number of proteomics datasets increases, systematic GO enrichment analyses will help address specific biological questions and provide new insights into understanding the etiology of muscle wasting. A multiple and integrative ‘-omics’ approach might also be used in the future, providing that data are generated from the same source material.

In our analysis, we deliberately excluded proteomics data from in vitro and animal models as, unlike studies carried out in humans, these studies tend to be more homogenous and controlled for, as well as published in greater numbers, therefore were likely to skew our GO analysis and the possible interpretation of the data generated.

A strength of our analysis is the ability to identify commonly affected biological processes and functions in different muscle atrophy conditions based on GO terms. This approach uses a tree structure where functionalities are related to each other through a hierarchical pyramid structure, reducing the impact of missing links or gaps in the individual data sets. It should be noted that across studies of cancer patients with or without cachexia, GO terms may derive from both cancer-associated and cachexia-associated proteins in the absence of validated biomarkers. Longitudinal studies following patients at diagnosis through to a cachectic stage may help identify which biomarkers are associated with cancer and its progression and which are genuinely cachectic markers.

In the future, an investigation of multifactorial syndromes such as muscle wasting will further require: (i) well-designed studies considering sex/hormonal status stratification, (ii) well-defined protocols for measuring patients’ global skeletal muscle status, complemented with robust quantitative methods for data analysis, and (iii) a panel of candidate molecular markers or biological pathways to leverage the current knowledge of disease onset and progression.

The first goal is bound to understand how the inherent differences at the anatomic and genomic levels of men and women impact the onset and progression of muscle wasting, contributing either by natural variation of sexual hormone homeostasis, the age of menopause onset or diseases that impair hormonal balance [75]. This challenge can, in part, be addressed with data from observational studies to infer the association of surgical treatment for gonad-related cancers with post-hormone replacement therapy (HRT) and the prevalence of muscle-wasting syndromes [82]. Additionally, balancing the proportion of men and women in newly designed clinical trials to accommodate for the sex-specific incidences of the primary disease in the study guarantees comparable statistical power [83].

To address the second goal, skeletal muscle status could be further inspected by employing creatine dilution tests throughout oral administration of deuterium-labeled creatine with a later assessment of the levels of D3-creatinine in urine. This method correlates well with MRI findings of whole-body muscle mass [84] and requires additional enhancement to be widely applied in the clinical setting. The incorporation of additional clinical measurements will also be important, such as ones borrowed from neuromuscular fields, including the number (MUNIX) and size (MUSIX) of motor neurons by electromyography which correlates with the degree of sarcopenia [85,86].

The development of physical exercise mimetics could prove to have stark benefits in the geriatric population with reduced mobility and/or in long-term hospitalization by minimizing loss of function and muscle mass throughout administration of these bioactive therapeutics [87]. Pre-clinical studies in mice have shown dual benefit of cardarine (GW1516), a peroxisome proliferator-activated receptor delta (PPARδ) agonist, together with exercise to increase running stamina [88]. In the same study, the long-term administration of 5-aminoimidazole-4-carboxamide ribonucleotide (AICAR) yielded an increase in running stamina. Recently, an in vitro cell culture study with human myofibers using interferon gamma (IFN-γ)-induced muscle wasting corroborated the use of tofacitinib and baricitinib (JAK/STAT inhibitors), two commonly prescribed therapeutics for rheumatoid arthritis (RA), which fully hindered muscle atrophy [89]. Additionally, neuromuscular electrical stimulation (NMES) which today lacks clinical adherence [90], could be used as a surrogate for physical exercise since it elicits skeletal muscle hypertrophy [91] and its efficacy is also currently under study with the joint administration of branched-chain amino acids (BCAA) in a cohort with liver transplantation [92].

In summary, our meta-analysis of individual proteomics studies provides an overview of uniquely and commonly affected biological processes and functions in various muscle atrophy conditions. This work demonstrates that GO analysis of published proteomics data can identify specific processes not revealed by single studies. Future efforts should focus on integrating data from proteomics analyses performed in various tissues (muscle, blood, serum/plasma, and/or urine) aiming at the comprehensive characterization and classification of the human skeletal proteome, to identify potential pathways and biomarkers of muscle wasting associated with various conditions.

## Figures and Tables

**Figure 1 molecules-27-05514-f001:**
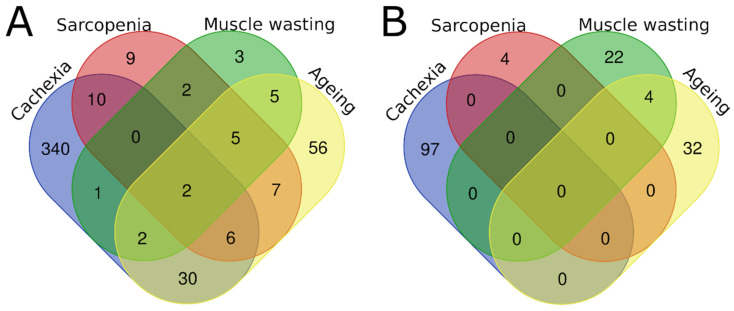
Venn diagram representing the number of overlapping proteins across all four muscle-wasting conditions identified in (**A**) tissue samples from 15 proteomics studies and in (**B**) biofluid samples from 10 proteomics studies.

**Figure 2 molecules-27-05514-f002:**
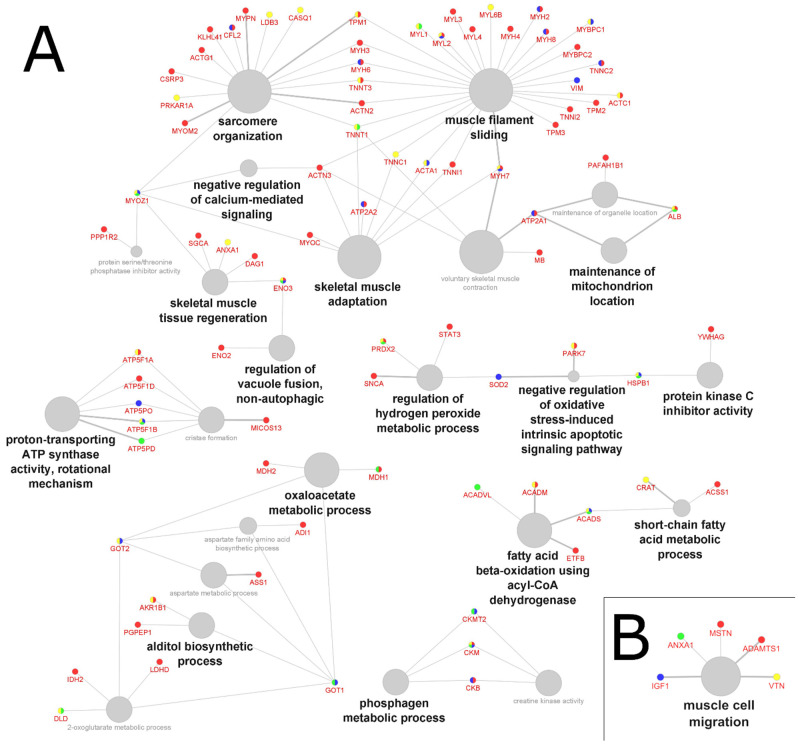
GO clustering based on logical “AND” for all four atrophy groups in (**A**) tissue samples and (**B**) biofluid samples from 24 proteomics studies. Grey nodes represent common (shared) up- and downregulated processes; the font and node sizes reflect statistical significance (kappa score = 0.4 and *p*-value < 0.01). The red, blue, green, and yellow colors associated with proteins arbitrarily represent the muscle atrophy groups.

**Figure 3 molecules-27-05514-f003:**
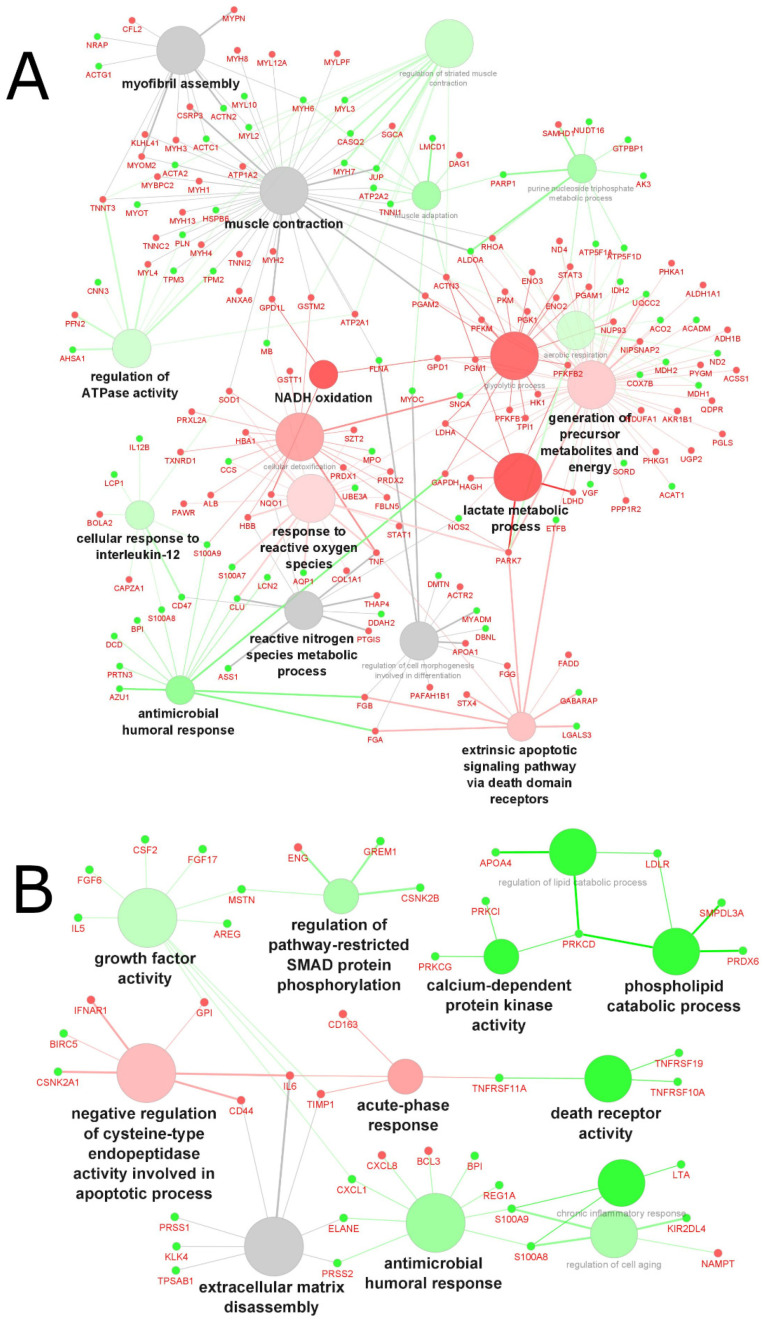
GO analysis of biological processes in cachexia-related muscle atrophy, based on grouped molecules identified in (**A**) tissue samples and (**B**) biofluid samples from 24 proteomics studies. Red nodes represent upregulated processes, green nodes represent downregulated processes, grey nodes represent common (shared) up- and downregulated processes; font and node sizes reflect statistical significance (kappa score = 0.4 and *p*-value < 0.01).

**Figure 4 molecules-27-05514-f004:**
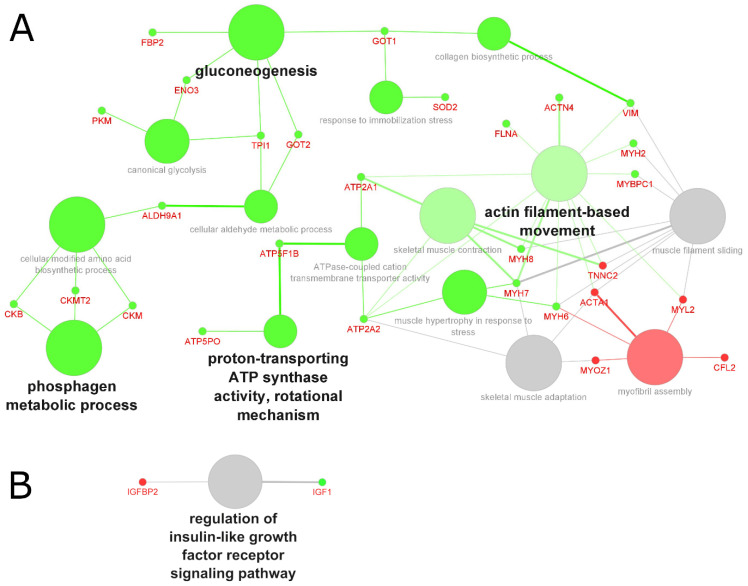
GO analysis of biological processes in sarcopenia-related muscle atrophy, based on grouped molecules identified in (**A**) tissue samples and (**B**) biofluid samples from 24 proteomics studies. Red nodes represent upregulated processes, green nodes represent downregulated processes, grey nodes represent common (shared) up- and downregulated processes; font and node sizes reflect statistical significance (kappa score = 0.4 and *p*-value < 0.01).

**Figure 5 molecules-27-05514-f005:**
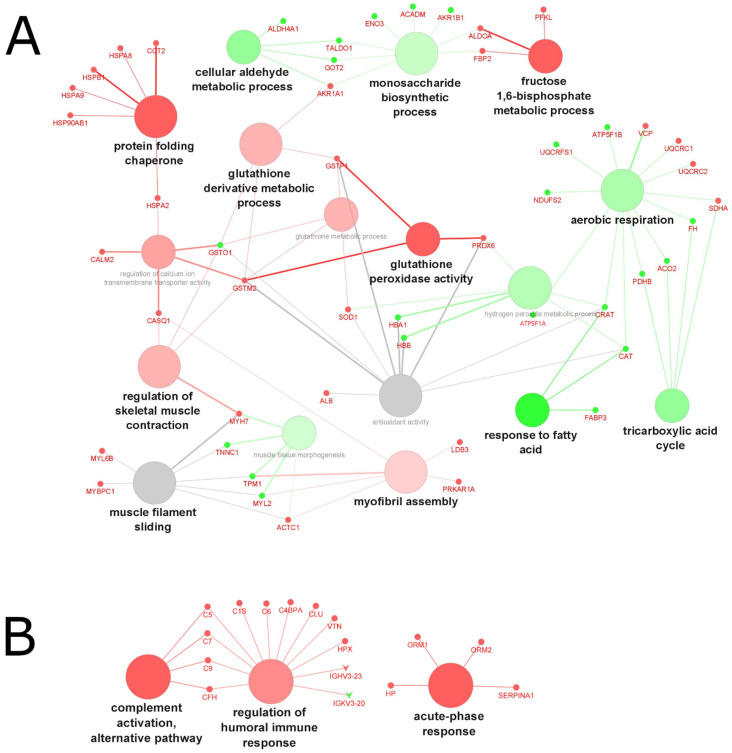
GO analysis of biological processes in aging-related muscle atrophy, based on grouped molecules identified in (**A**) tissue samples and (**B**) biofluid samples from 24 proteomics studies. Red nodes represent upregulated processes, green nodes represent downregulated processes, grey nodes represent common (shared) up- and downregulated processes; font and node sizes reflect statistical significance (kappa score = 0.4 and *p*-value < 0.01).

**Figure 6 molecules-27-05514-f006:**
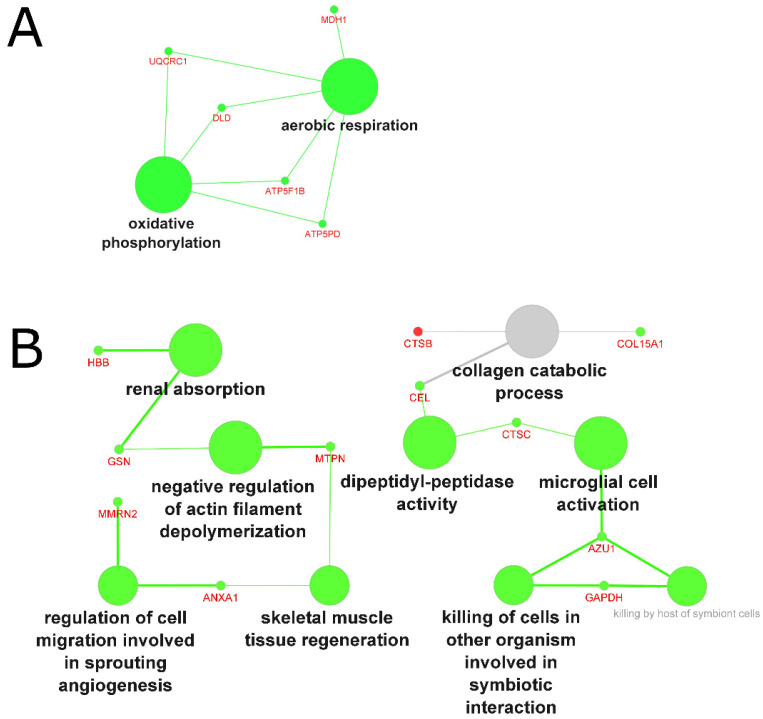
GO analysis of biological processes in muscle wasting-related atrophy, based on grouped molecules identified in (**A**) tissue samples and (**B**) biofluid samples from 24 proteomics studies. Green nodes represent downregulated processes, grey nodes represent common (shared) up- and downregulated processes; font and node sizes reflect statistical significance (kappa score = 0.4 and *p*-value < 0.01).

**Figure 7 molecules-27-05514-f007:**
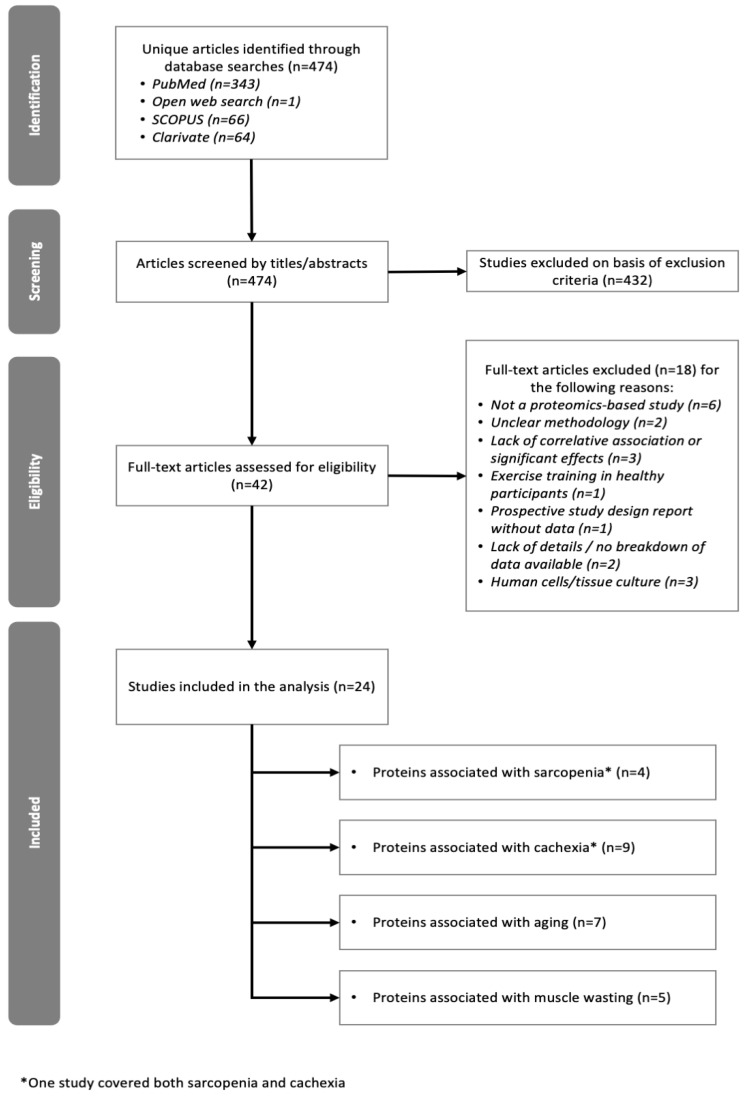
Flowchart of methodology used to identify studies included in the meta-analysis.

**Table 1 molecules-27-05514-t001:** Study details of articles included.

Study	Muscle Atrophy Condition	N	Groups	Tissue Analyzed
Ebhardt et al. (2017) [21]	Cachexia	19	Cachectic (*n* = 5) Non-cachectic (*n* = 14)	*Quadriceps* muscle biopsy
Zhou et al. (2020) [22]	Cachexia	23	Cachexia with sarcopenia (*n* = 13) Normal weight (*n* = 10)	Abdominal muscle biopsies
Aniort et al. (2019) [23]	Cachexia	21	Early-stage lung cancer (LC, *n* = 7) Chronic hemodialysis (HD, *n* = 7) Healthy volunteers (CT, *n* = 7)	Muscle biopsies (LC: *latissimus dorsi*; HD and CT: *vastus lateralis*)
Costa et al. (2019) [24]	Cachexia	45	Cachectic (*n* = 25) Weight stable (*n* = 20)	Plasma
Narasimhan et al. (2020) [25]	Cachexia	29	Weight loss (*n* = 23) Weight stable (*n* = 6)	Serum
Neto et al. (2018) [26]	Cachexia	16	Cachectic (*n* = 9) Weight stable (*n* = 7)	Peritumoral adipose tissue
Muqaku et al. (2017) [27]	Cachexia	9	Melanoma (*n* = 6) Healthy (*n* = 3)	Serum
Skipworth et al. (2010) [28]	Cachexia	16	Weight-stable (*n* = 8) Weight-loss (*n* = 8)	Urine
Arner et al. (2015) [29]	Cachexia	59	Weight-stable (*n* = 27) Weight-loss (*n* = 32)	Plasma
Ebhardt et al. (2017) [21]	Sarcopenia	18	Sarcopenic (*n* = 8) Non-sarcopenic (*n* = 10)	*Quadriceps* muscle biopsy
Gueugneau et al. (2021) [30]	Sarcopenia (metabolic syndrome)/aging	39	Healthy young (*n* = 15) Healthy old (*n* = 15) Old with metabolic syndrome (*n* = 9)	*Vastus lateralis* muscle biopsy
Bergen et al. (2015) [31]	Sarcopenia	240	Young (20–40y, *n* = 80) Old with normal rASM (>65y, *n* = 80) Old with low rASM (>65y, *n* = 80)	Serum
L’hôte et al. (2021) [32]	Sarcopenia	20	Control and pre-sarcopenia (*n* = 10) Sarcopenia and severe sarcopenia (*n* = 10)	Serum
Lin et al. (2017) [33]	Frailty/ Aging	12	Frail (*n* = 6) Non-frail (*n* = 6)	Serum
Brocca et al. (2017) [34]	Aging	20	Elderly (70.9y, *n* = 10) Young control (23.0y, *n* = 10)	*Vastus lateralis* muscle biopsies
Ubaida-Mohien et al. (2019) [35]	Aging	58	20–34 years (*n* = 13) 35–49 years (*n* = 11) 50–64 years (*n* = 12) 65–79 years (*n* = 12) 80+ years (*n* = 10)	*Vastus lateralis* muscle biopsy
Lourenço Dos Santos et al. (2015) [36]	Aging	22	Young healthy (0–12y, *n* = 11) Old healthy (52–76y, *n* = 11)	*Rectus abdominis* muscle biopsies
Gueugneau et al. (2014) [37]	Aging	24	Mature women (48–61y, *n* = 11) Older women (76–82y, *n* = 13)	*Vastus lateralis* muscle
Théron et al. (2014) [38]	Aging	10	Mature healthy (53.0y, *n* = 6) Old healthy (77.6y, *n* = 4)	*Vastus lateralis* muscle
Staunton et al. (2012) [39]	Aging	8	Middle aged (47–62y, *n* = 4) Older (76–82y, *n* = 4)	*Vastus lateralis* muscle biopsies
Rittweger et al. (2018) [40]	Muscle wasting	2	Crew members of the ISS assessed pre and post 6 month stay in space	Skeletal (*soleus*) muscle
Capri et al. (2019) [41]	Muscle wasting	2	Crew members of the ISS assessed pre and post 6 month stay in space	Muscle biopsy
Husi et al. (2018) [42]	Muscle wasting	49	Low strength (22/49) Low power (23/42) Low strength and power (*n* = 13)	Urine
Lakhdar et al. (2017) [43]	Muscle wasting	27	COPD with low FFMI, (*n* = 10) COPD with normal FFMI (*n* = 8) Matched healthy controls (*n* = 9)	*Vastus lateralis* muscle of COPD patients
Husi et al. (2018) [44]	Muscle wasting	55	Myosteatotic (*n* = 31) Non-myosteatotic (*n* = 24)	Urine

COPD, chronic obstructive pulmonary disease; FFMI, fat free mass index; ISS, International Space Station; rASM, relative appendicular skeletal muscle mass; y, years.

**Table 2 molecules-27-05514-t002:** Overlap of proteins found up- and/or downregulated across studies, per muscle condition and tissue or biofluid samples.

	Cachexia		Sarcopenia		Muscle Wasting		Aging	
	Tissue *	Biofluid †	Tissue *	Biofluid †	Tissue *	Biofluid †	Tissue *	Biofluid †
**Number of studies**	4	5	2	2	3	2	6	1
**Total number of proteins**	391	97	41	4	20	26	113	37
**Number of proteins overlapping across studies**	21	1	3	0	0	1	34	NA
**Number of overlapping proteins with same directionality**	17	0	3	0	0	1	15	NA

* Muscle biopsies; † Serum, plasma, or urine. NA, not applicable.

**Table 3 molecules-27-05514-t003:** List of overlapping proteins identified with each muscle-wasting condition.

Atrophy Condition	Proteins in Tissue Samples *	Proteins in Biofluid Samples *
**Cachexia**	S100A8, ENO3, PKM, MYH6, ATP2A1, TNNC2, GAPDH, ACTA2, TPM1, PGK1, ANXA6, PFKM, AK1, MYBPC2, TPD52L2, ACTBL2, SERBP1, CDS2, STT3B, GMPR, FBLN5 (21/391)	SPTAN1 (1/97)
**Sarcopenia**	HSPB1, GOT1, MYL2 (3/41)	– (0/4)
**Muscle wasting**	– (0/20)	GFAP (1/26)
**Aging**	CKM, PYGM, CA3, ACTA1, HSPB6, TNNT3, ANXA5, TNNT1, MYL1, ANKRD2, GAPDH, PKM, ENO3, MYL2, ACTC1, ALDOA, PRDX2, MYH1, LDHB, TPI1, PGM1, PARK7, GPD1, MYOZ1, ATP5B, DLDH, CRYAB, COX5A, PRDX3, ALDH2, FH, TTN, FABP4, TF (34/113)	– (0/37)

* Numbers in bracket denote the number of proteins that overlap across studies, out of the total number of proteins identified in the samples.

## Data Availability

The data presented in this study are available in Appendix A.

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
