# Peer review of "Gene Ontology (GO)-Driven Inference of Candidate Proteomic Markers Associated with Muscle Atrophy Conditions"

_molecules, 2022, doi:10.3390/molecules27175514_

Round 1

Reviewer 1 Report

The work of Stalmach et al. is a good example of integrative analysis of massive data focused on answering questions that need to be bridged to cover the gaps left by individual studies. The work has been carried out adequately, but I think it would be very useful to include data from each study regarding the methods of protein extraction and sample processing; the type of approach used, whether label-free, isobaric labelling, absolute or relative quantification, discovery, or targeted studies; the type of mass spectrometry platform used. Taking these factors into consideration, it may be possible to broaden the discussion regarding the low coincidence observed in terms of the number and type of proteins between studies.

Author Response

We would like to thank the Reviewer for their comments and suggestions. We provided (in supplementary Table S1) additional details on some aspects of the methodology described in each individual study, with regard to protein extraction and sample processing, type of analysis, and quantitation method used (see additional information in columns I, J, and K in supplementary Table S1). We also discussed the possibility that these factors may contribute to the low number of overlapping proteins observed across studies in the body of text (see track changes, section 2.2, p. 6-7) and summarised below:

‘In our current analysis, possible factors that might explain the low proportion of overlapping proteins across studies include the many types of cancer (6 different types across 9 studies), and heterogenous definition of cachexia in the individual studies. Similarly, heterogeneity in the definition of weight loss and patient characteristics across studies might partly explain the low proportion of overlapping proteins with the same directionality in sarcopenia, muscle wasting and aging-related muscle atrophy condition. Additionally, the different methodologies observed across studies in relation to protein extraction and sample processing, analytical and quantitation approaches, and mass spectrometry platform (see in Supplementary Table S1) may lead to the identification of a wide range of different proteins. Altogether, the lack of a standardized approach for generating proteomics data may contribute to a low overlap of molecules across studies.’ (this paragraph was amended and moved from section 2.4.1, p. 10 to section 2.2, p. 6-7, as identified by the authors to be a more relevant section for discussing factors that contribute to the low number of overlapping proteins observed across studies).

Reviewer 2 Report

The manuscript sounds relevant and brings interesting analysis on muscle atrophy conditions, integrating proteomic data through gene ontology approach. The manuscript is well written, the results are very consistent and the discussion is very complete, but has some minor issues that need revision and/or correction before publication.

MINOR:

Abstract: I suggest to indicate in the 11th line which would be the main common biological processes. The Abstract section should better specify the results obtained in the study.

Introduction: Specify in the Introduction section the relationship between skeletal muscle, adipose tissue and the biofluids. This is important to justify why the authors did not evaluate only the muscle tissue studies.

Results:

Page 5, 1st paragraph: Change the sentence “protein level across the four muscle-wasting conditions” to “protein level across the four muscle atrophy conditions”, to avoid confusion.

Page 6, subtopic 2.3.: Discuss these common enriched terms and their relevance to skeletal muscle atrophy a little further, adding some references. In addition, I believe it is important to indicate in the text (not just in the figure) which of these functions was most enriched.

Supplementary Figure S1A: I was not able to open the figure S1. I believe it was a problem with the program on my computer, but it is better to check if everything is ok.

Figure 2: Considering that Figure 2 does not show red or green nodes related to the processes, delete this part: “Red nodes represent up-regulated processes, green nodes represent down-regulated processes”. However, this sentence is necessary in the Figure 3 legend. Also, what do the colors for each protein (red, blue, green, yellow) mean? Some of them have more than one color.

Page 10, 3rd to 6th paragraphs: I suggest to summarize this part. These paragraphs should contain more discussion about the results of this study, but they seem more related to other works.

Figure 5 legend: Correct to “processes in aging-related muscle atrophy”.

Page 14, 2nd paragraph: I also suggest to summarize this part.

Conclusions: Conclusion section focused mainly on the future requirements to conduct the muscle atrophy studies, but lacks information on the processes/functions identified in the present study and the importance of these results.

Author Response

We would like to thank the Reviewer for their comments and recommendations. We have made revisions and corrections as suggested and provided clarification where required. Please see our responses to the Reviewer’s comments below, detailed point by point.

Abstract: I suggest to indicate in the 11th line which would be the main common biological processes. The Abstract section should better specify the results obtained in the study.

Response: Although we agree with the Reviewer’s suggestion, the abstract as is currently written has reached the maximum word count limit (200 words). We feel that a clearer summary of the main common biological processes as well as differences observed are best placed in the conclusion section, as was also suggested by the Reviewer in a later comment.

Introduction: Specify in the Introduction section the relationship between skeletal muscle, adipose tissue and the biofluids. This is important to justify why the authors did not evaluate only the muscle tissue studies.

Response: We have amended the introduction to include details that highlight the relationship between skeletal muscle, adipose tissue and the biofluids and the importance to evaluate tissue proteomics beyond skeletal muscle, as suggested (see track changes, section 1., p. 1-3 and summarised below):

‘Skeletal muscle amounts to ~40% of the human body weight and entails 50–75% of all body proteins [1,2]. Moreover, skeletal muscle tissue is of complex tissue architecture consisting of a multitude of cell types: immune cells, endothelial cells, fibroadipogenic progenitors (FAPs), tenocytes, satellite cells, muscle fibers, and neural cells [3]. These cells communicate amongst each other in a paracrine fashion, and skeletal muscle itself has also been shown to communicate to other organs and tissues (i.e. adipose tissue) in an endocrine fashion through secretion of so-called myokines by muscle contraction [4]. Conversely, adipose tissue may also play an active role in cancer-associated cachexia, whereby lipolysis has been shown to be permissive of skeletal muscle wasting and an es-sential part in the pathogenesis of cancer-associated cachexia [5].’

‘Therefore, the knowledge that skeletal muscle tissue plays a significant role in endocrine signaling, in particular upon insult, warrants the study of protein biomarkers in not only skeletal muscle, but also tissue affected by secretion of such myokines: biofluids.

The capacity of skeletal muscle to secrete myokines gives biofluids enormous prognostic power since proteomic biomarkers could potentially predict disease states at timepoints shortly after secretion, before the body had enough time to enact physiological change. Likewise, the known para- and endocrine role that adipo- and myokines play in the regulation of each other could prove useful in biomarker discovery [20]. One should note that osteokines also strongly contribute to this signaling triad between muscle, adipose, and bone, however, does not easily lend itself to biopsies.’

Results:

Page 5, 1st paragraph: Change the sentence “protein level across the four muscle-wasting conditions” to “protein level across the four muscle atrophy conditions”, to avoid confusion.

Response: We have amended the sentence as suggested (see track changes, section 2.2, p. 5).

Page 6, subtopic 2.3.: Discuss these common enriched terms and their relevance to skeletal muscle atrophy a little further, adding some references. In addition, I believe it is important to indicate in the text (not just in the figure) which of these functions was most enriched.

Response: We do agree that the current section benefits from additional details about the common biological processes and functions identified in our analysis. We elaborated further and added some references to support our findings (see paragraph in track changes, section 2.3, p. 7 and summarised below):

‘The convergent biological processes and function across all four atrophy conditions center around seven protein-driven GO clusters that can be described as skeletal mass and function, oxidative stress response, energy transfer, phosphagen system, glycolytic system, and oxidative (mitochondrial) system (tissue samples, Figure 2A), and muscle tissue repair (biofluid samples, Figure 2B). Indeed, different physio-pathologies that result in muscle atrophy, such as age-related sarcopenia and cancer-induced cachexia, have been shown to exhibit similar molecular, cellular, and systemic effects characterized by impaired protein homeostasis, metabolic remodeling, impaired mitochondrial function, increased oxidative stress response, impaired energy expenditure, and altered muscle composition leading to muscle wasting [45-48]. Our analysis therefore shows that changes in muscle adaptation and metabolic processes might be an indicator of muscle damage and/or remodeling underlying all four atrophy conditions. Muscle wasting conditions present similarities in their clinical manifestations and can occur concurrently in the same patient [49]; however, they do have distinct etiologies, and factors that promote cachexia are known to be different from those involved in sarcopenia with inflammation being one of the key features in cachexia [50-52]. This suggests that specific protein-driven GO terms may be associated with each muscle atrophy condition. We next examined the proteomic changes associated with atrophy induced by cachexia, sarcopenia, aging, and muscle wasting. Our analysis suggested that distinguishable biological processes and functions were affected in the different conditions under study, as summarized in the subsequent sections.’

Supplementary Figure S1A: I was not able to open the figure S1. I believe it was a problem with the program on my computer, but it is better to check if everything is ok.

Response: Many thanks for pointing out. We have converted the figure into a JPG file and checked that it could open.

Figure 2: Considering that Figure 2 does not show red or green nodes related to the processes, delete this part: “Red nodes represent up-regulated processes, green nodes represent down-regulated processes”. However, this sentence is necessary in the Figure 3 legend. Also, what do the colors for each protein (red, blue, green, yellow) mean? Some of them have more than one color.

Response: We have deleted the irrelevant part of the sentence, as per suggestion.

The red, blue, green, and yellow colours represent each group (muscle wasting, aging, sarcopenia, and cachexia). However, the colour coding itself does not have any bearing on the analysis or interpretation. We have none-the-less included a clarification in the legend of Figure 2, as follows: ‘The red, blue, green, and yellow colors associated with proteins arbitrarily represent the muscle atrophy groups’.

Page 10, 3rd to 6th paragraphs: I suggest to summarize this part. These paragraphs should contain more discussion about the results of this study, but they seem more related to other works.

Response: In these paragraphs, we included and discussed elements of individual studies that were included in our analysis, in order to highlight potential biomarkers and pathways of interest. We have made clearer that the discussion pertained to the studies included in the analysis (see track changes, p. 11: ‘…several studies (10 out of 24) have profiled…’. We also removed part of the discussion that was not relevant to our current analysis in order to summarise this section, as per suggestion (see removed text in track changes, p. 11: ‘CNDP1 is a metalloprotease that catalyzes the degradation of carnosine. Although its function is not fully known in disease, altered activity has been associated with several pathological conditions [36]. Reduced levels have been found in serum samples of patients with Duchenne muscular dystrophy compared with healthy controls.’)

Figure 5 legend: Correct to “processes in aging-related muscle atrophy”.

Response: Many thanks for pointing out. We have corrected the figure legend accordingly.

Page 14, 2nd paragraph: I also suggest to summarize this part.

Response: We have revised this section into a more concise paragraph, by deleting the following sentences (see track changes, section 2.4.3, p. 15):

‘Among the study participants, none of the premenopausal women used hormonal therapy, while the postmenopausal women were monozygotic co-twin pairs with one sister being a current HRT user while the other never used HRT.’

‘Furthermore, 17β-estradiol was predicted to be an upstream regulator of these processes (confirmed by in vitro exposure of myotubes to 17β-estradiol) and several 17β-estradiol-associated differentially expressed proteins were identified for each group comparison.’

‘Although muscle atrophy is a key feature of sarcopenia, dynapenia has negative consequences, with faster rates of decreases in grip and hip flexor strengths in women being independent predictors of mortality.’

Conclusions: Conclusion section focused mainly on the future requirements to conduct the muscle atrophy studies, but lacks information on the processes/functions identified in the present study and the importance of these results.

Response: We agree with the Reviewer’s suggestion, and included a summary of our current findings, as well as highlighting the importance of the results (see track changes, section 3., p. 17, and summarised below):

‘Common biological processes and functions across all four atrophy conditions pertained to skeletal mass and function, oxidative stress response, energy transfer, phosphagen / glycolytic / oxidative systems, and muscle tissue repair. Changes in muscle adaptation and metabolic processes might therefore be mechanisms associated with muscle damage and/or remodeling common to all four atrophy conditions. Additionally, our analysis identified distinct biological processes and functions associated with each muscle atrophy condition. The major biological processes and functions impacted in cachexia appeared to largely relate to changes in host metabolism, with lactate metabolic process, NADH oxidation, and glycolytic process identified as the major enriched pathways for the upregulated genes, presumably driven by a metabolic reprogramming of host cells via host-tumor cell crosstalk. By contrast, key changes associated with sarcopenia included downregulation of pathways related to energy metabolism, indicative of a preferential atrophy of fast-twitch fibers and glycolytic-to-oxidative metabolic shift that could contribute to the loss of muscle force. In aging, upregulation of protein folding chaperone, muscle contraction, energy metabolism, and cellular oxidative activities, together with an impaired mitochondrial capacity observed in tissue biopsies of older vs younger individuals are known hallmarks of cellular senescence promoting loss of protein quality control, thereby preventing repair of protein damage, leading to degeneration and cell death. Similar to the aging process, muscle wasting was associated with mitochondrial dysfunction, and impaired cell activation, migration, and regeneration, possibly indicative of an impaired cellular anabolism. Altogether, our analysis revealed common and specific biological processes and functions across various muscle atrophy conditions which, despite their distinct etiologies, often present with similar manifestations and poor patient outcomes. At the molecular level, each of the muscle atrophy conditions is distinguishable from one another, driven by different progenitor events, which brings the possibility to develop targeted treatment and improve overall outcomes.’

Round 2

Reviewer 1 Report

The recommendations suggested in the first round of revision have been addressed. No comments.